

# Automatic True North detection during absolute magnetic declination measurement

Alexandre Gonsette[1], Jean Rasson[1], Stephan Bracke[1], Antoine Poncelet[1], Olivier Hendrickx[1], François Humbled[1]

[1]Dourbes magnetic observatory, Royal Meteorological Institute of Belgium, Dourbes, 5670, Belgium

*Correspondence to*: Alexandre Gonsette (agonsett@meteo.be)

**Abstract.** Absolute magnetic measurements are of great importance in magnetic observatories. They allow not only instrument calibration but also data quality checking. They require the Vertical and the geographic or True North as reference directions, usually determined by means of a level and by pointing an azimuth mark respectively. We present here a novel system able to measure the direction of the magnetic field, of the Vertical and True-North. A design of a North-seeker is proposed taking into account sensor bias as well as misalignment errors. Different methods are derived from this model and measurement results are presented. A measurement test at high latitude is also shown.

## 1 Introduction

Measuring the magnetic declination is realized by determining, in a horizontal plane, both magnetic field and geographic or True North direction (In the rest of this paper, the term True North will be employed). Then the angle between them is computed. In magnetic observatory as well as in the field, this value is measured by an observer during the so-called "absolute" measurement step (Rasson, 2005). This procedure consists of two main steps in manipulating a DIFlux. First, the instrument is oriented relative to the magnetic field in order to establish its direction in space. Practically, the magnetic field sensor mounted on the telescope is placed in the horizontal plane. The sensor output is then the projection of the field horizontal component along the sensor sensitive axis or in other words, the scalar product of both. The most sensitive direction is therefore perpendicular to the magnetic field. Then, the True North is determined by pointing at a target whose azimuth is already known. Finally the observer extracts the magnetic declination from both readings. The target azimuth can be established by different methods: by a gyro-theodolite; by pointing at a celestial body such as the Sun in combination with a clock; or by using a GNSS system (Newitt et al., 1996). In any case, this target azimuth value is measured prior to the declination measurement and is assumed constant until it is checked again.

During the last years, efforts have been made in order to automatize absolute magnetic measurements. Niemegk observatory developed the GAUSS based on a 3-axis fluxgate sensor rotating sequentially around two known directions (Auster et al., 2007). At the same time, Dourbes observatory successfully attempted to robotize the DIFlux absolute measurement



procedure leading to the AutoDIF instrument (Rasson and Gonsette, 2011). Today a few of them are operational in different observatories. However, both GAUSS and AutoDIF use the target pointing principle for the True North measurement. The development of an automatic observatory will allow its deployment in remote areas but consequently raises new challenges that were not considered up to now. What would happen if no target is available or if it is not stable? Arctic regions are good

candidates to host autonomous systems (Marsal et al., 2017) but drifting ice or permafrost requires a constant azimuth update (Eckstaller et al., 2007). Furthermore the idea of automatic observatories also calls for a need for automatic True North direction determination. The system described in this paper is an automated DIFlux based on Autodif architecture in which the target pointing system has been replaced by an embedded True North seeker.

## 2 Background

A Fiber Optical Gyroscope (FOG) is an absolute rotation sensor and may be able to detect the Earth rotation. Its principle is based on the Sagnac effect (Arditty and Lefèvre, 1981). Without entering too much in details, let us just imagine two balls rolling at the same speed but in opposite direction at the circumference of a disc. If the disc is static, an external observer would see both balls crossing each other after half a turn and again at the start point. If the disc is put into rotation, the balls will not reach the start point relative to an inertial frame at the same time. The delay is therefore proportional to the disc

rotation speed. FOG-based sensors use a similar principle: two light beams traveling at the same speed along an optical fiber are injected from each end. The phase shift between the two optical waves gives the sensor rotation speed.

### 2.1 Static method

North seeker methods are usually sorted in two categories: Static (Liu et al., 2014) and dynamic (Xu and Guo, 2010). In both cases, the sensitive axis of the FOG is directed horizontally and the projection of the Earth rotation vector on it is given by

$$\omega(\phi) = \Omega_e \cos(\theta) \cos(\phi) + b \tag{1}$$

With

- $\omega$     Angular speed recorded by FOG
- $\phi$     Angle between True North and the direction pointed by the FOG's sensitive axis
- $\theta$     Latitude
- $\Omega_e$     Earth rotation speed $\sim 15.041 \frac{°}{h}$
- $b$     FOG bias

In the static method, two opposite directions are pointed in order to compensate for the bias. Due to the $\cos(\phi)$ term, the most sensitive directions lie along the East-West axis. The True-North is then found by adding or removing 90 degrees from

the result. Additional pointing close to the East-West may be required so that the FOG sensor scale factor can be calibrated. For automatization purpose, it is possible to point at still more directions in order to remove the East-West uncertainty.

$$\omega_1(\phi) = \Omega_e \cos(\theta) \cos(\phi) + b$$

$$\omega_2(\phi + \pi) = \Omega_e \cos(\theta) \cos(\phi + \pi) + b \qquad (2)$$

$$\phi = \text{acos}\left(\frac{\omega_1 - \omega_2}{2\Omega_e \cos(\theta)}\right) + w_n$$

Where $w_n$ is an instrumental white noise. The previous equation suggests to increase the sampling time in order to reduce the white noise according to central limit theorem. It is not exactly true because the bias is subject to drift due to environmental changes like temperature. The problem is therefore to find the optimum sampling time that minimize both white noise and drift contribution to uncertainty. The Allan variance is a useful tool for this. Since the FOG remains stationary during each acquisition step its output is supposed to remain constant. The minimum in the curve of the Allan variance gives the
optimum acquisition time as well as the uncertainty level of measurement.

## 2.2 Dynamic and combined method

In the dynamic method, the FOG's sensitive axis is also kept horizontal but continuously turns around a vertical axis. The phase shift of the FOG output gives the True North direction ($\pm 90°$)with respect to the arbitrary zero of the angle reading in the instrument frame. This method is not affected by the sensor bias so that at first glance it could be preferred to the static
one. Unfortunately the sensitivity of FOG sensors is too low to allow this dynamic method to be used for azimuth determination in the particular case of magnetic declination measurement.

It is also possible to combine both methods by performing static measurements at regular spaced angular positions (Abbas, 2013). In this hybrid case, the sampling time can be optimum. The output is therefore a discrete sinus curve whose amplitude
is given by $\Omega_e \cos(\theta)$. The phase shift gives the True North direction ($\pm 90°$).

## 3 New approach

The above True North methods do not consider a possible FOG misalignment. However, it is evident that a horizontal misalignment has a direct impact on the measurement. Again, since the sensor is supposed to measure the horizontal component of the earth rotation vector (see Eq. 1), a vertical misalignment also leads to an error due to the orthogonal
projection of vertical component of the earth rotation vector onto the plane of measurement of the FOG sensor. Many FOG-based North-seekers only have the possibility to rotate around the vertical axis so that they do not have the opportunity to take misalignment into account. When looking to the accuracy of magnetic declination required by international standards like those established by INTERMAGNET (Intermagnet, 2012), it appears evident that such error must be compensated.





Indeed, the 5nT maximum allowed error on the Y component leads to a maximum misalignment error (case in Dourbes with $H_m = 20\mu T$):

$$Misalignment\ error = \frac{180}{\pi}\text{atan}\left(\frac{5}{20000}\right) = 0.014°, \qquad (3)$$

Reciprocally, a small 0.05° misalignment error would correspond to 17.5nT.

## 3.1 Gyro-DIFlux

Because a DIFlux has two principal degrees of freedom, a FOG sensor mounted in the same reference frame as magnetic sensor (i.e. on the telescope in the case of conventional DIFlux such as Zeiss 020) can be oriented in any direction in space. Moreover, the FOG magnetic signature participates to the magnetic sensor offset and is compensated by the declination/inclination measurement protocol (Gilbert and Rasson, 1998).

The GyroDIF instrument is a non-magnetic robotized platform able to orient sensors in any direction. It is based on the AutoDIF system. A fluxgate sensor and a FOG are mounted on the horizontal axis. Neglecting misalignment errors, both have their sensitive direction parallel. Piezoelectric motors can rotate the horizontal and vertical axes with a resolution up to 0.001°. An electrolytic level continuously records tilt errors with 0.1 arcsec resolution and 1arcsec linearity. Non-magnetic angular encoders allow angles measurement with accuracy better than 6 arcsec according to ISO 17-123 (Gonsette et al., 2014). The angle readings respect the DIFlux conventions with a horizontal circle increasing clockwise and a vertical circle such that zero is read when fluxgate is roughly vertical and +90° when fluxgate is horizontal on top of the axis (commonly sensor UP). Figure 1 presents the GyroDIF implementation.

## 3.2 An extended model

In the middle of the 1980ies Kring Lauridsen (Lauridsen, 1985) and David Kerridge (Kerridge, 1988) established a model mathematically describing the magnetic field vector in the DIFlux sensor reference frame. The theodolite was supposed to have two degrees of freedom, perfectly leveled and free of mechanical errors such as orthogonality errors or play in axes. They included a sensor offset and two angles describing the misalignment of the fluxgate sensitive axis relative to the telescope optical axis. Kerridge model leads to the following equation:

$$T = Hcos(\phi - D)(\cos(\beta) - \epsilon\sin(\beta)) - \gamma Hsin(\phi - D) + Z\big(sin(\beta) + \epsilon cos(\beta)\big) + T_0, \qquad (4)$$

Where $H, Z, D$ are the geomagnetic horizontal, vertical and declination components respectively; $\epsilon, \gamma$ are the vertical and horizontal sensor misalignments respectively; $T, T_0$ are the sensor output and offset respectively; $\phi, \beta$ are the rotation angles around the vertical and horizontal axes relative to True North and horizontal respectively. From the previous equation,



Kerridge derived a method based on 4 measurements leading to a final declination measurement result free of those three errors (at first order). Similar development has been performed for magnetic inclination.

However considering a platform like the GyroDIF with two orthogonal rotation axes, a similar model can be implemented. Furthermore this system also records its tilt angle that could be modeled by two angular degrees of freedom. The earth rotation vector can be expressed in the FOG sensor reference frame with Z axis in the sensor axis direction and considering small tilt and misalignment angles:

$$\vec{\omega} = \begin{bmatrix} 1 & 0 & -\epsilon_g \\ 0 & 1 & \gamma_g \\ \epsilon_g & -\gamma_g & 1 \end{bmatrix} R_y(\beta) R_x(\phi) \begin{bmatrix} 1 & B & -A \\ -B & 1 & 0 \\ A & 0 & 1 \end{bmatrix} \begin{bmatrix} cos(\theta) & 0 & -sin(\theta) \\ 0 & 1 & 0 \\ sin(\theta) & 0 & cos(\theta) \end{bmatrix} \begin{bmatrix} 0 \\ 0 \\ \Omega_e \end{bmatrix} + \begin{bmatrix} T_x \\ T_y \\ T_z \end{bmatrix}, \tag{5}$$

With

- $T_{xyz}$     Sensor offset in the X,Y,Z direction
- $A, B$     Northward and Eastward tilt angles
- $R_{x,y}(X)$ Elementary rotation matrix around a local X,Y axis
- $\phi$     Angle between True North and sensor axis (neglecting misalignment angles)
- $\beta$     Angle between Horizontal plan and sensor axis (neglecting misalignment angles)
- $\epsilon_g$     FOG misalignment in the vertical plane
- $\gamma_g$     FOG misalignment in the horizontal plane

Considering the GyroDIF as shown in the Fig. 1, the FOG output is given by computing the third component of previous equation $\omega_3$. The similitude with Eq. (4) derived from Kerridge model is evident. Only the leveling terms are added.

$$\omega_3 \approx H_e \cos(\phi)\big(\cos(\beta) - \epsilon_g \sin(\beta)\big) - \gamma_g H_e \sin(\phi)$$

$$-Z_e\big(\sin(\beta) + \epsilon_g \cos(\beta)\big) - Z_e \cos(\beta)(A\cos(\phi) + B\sin(\phi)) + T_z, \tag{6}$$

Where $H_e = \Omega_e \cos(\theta)$ and $Z_e = \Omega_e \sin(\theta)$.

### 3.3 Four positions method

The static method can be adapted in order to compensate for the FOG misalignment. For an arbitrary direction $\phi$, Eq. (6) leads to four equations. Small angles approximations are made for $\beta \approx 0$ and $\beta \approx \pi$:

$$\omega_{3a}(\phi, \beta = 0) \approx H_e \cos(\phi) - \gamma_g H_e \sin(\phi) - Z_e(\beta_a + \epsilon_g) - Z_e(A\cos(\phi) + B\sin(\phi)) + T_z, \tag{7}$$

$$\omega_{3b}(\phi, \beta = \pi) \approx -H_e \cos(\phi) - \gamma_g H_e \sin(\phi) + Z_e(\beta_b + \epsilon_g) + Z_e(A\cos(\phi) + B\sin(\phi)) + T_z, \tag{8}$$




$$\omega_{3c}(\phi + \pi, \beta = \pi) \approx H_e \cos(\phi) + \gamma_g H_e \sin(\phi) + Z_e(\beta_c + \epsilon_g) - Z_e(A \cos(\phi) + B \sin(\phi)) + T_z , \qquad (9)$$

$$\omega_{3d}(\phi + \pi, \beta = 0) \approx -H_e \cos(\phi) + \gamma_g H_e \sin(\phi) - Z_e(\beta_d + \epsilon_g) + Z_e(A \cos(\phi) + B \sin(\phi)) + T_z , \qquad (10)$$

Combining Eq (7) to (10), almost all errors vanish at first order. It is reasonable to consider the horizontality errors $Z_e \beta$ as random errors that also vanish while the number of measurements increases. The resulting angular speed is:

$$\omega_r = \frac{\omega_a - \omega_b + \omega_c - \omega_d}{4} \approx H_e \cos(\phi) - Z_e(A \cos(\phi) + B \sin(\phi)) , \qquad (11)$$

The last term corresponds to the levelling error monitored by the electronic level. The angle relative to True North is then given by:

$$\phi = a\cos\left(\frac{\omega_r}{H_e} + \tan(\theta)(A \cos(\phi) + B \sin(\phi))\right), \qquad (12)$$

From Eq. (12), the optimum measurement direction is the east-west axis ($\phi \approx 90°$). In this case, the resulting angular speed is close to zero in the quasi-linear part of the cosine function. However, Eq. (12) does not take into account a possible scale factor uncertainty. The sensor output is usually a voltage or a digital value that need to be converted in corresponding angular speed. An error in $\omega_r$ introduces an error in the True-North determination. To reduce this effect a solution consists of performing two sets of 4 measurements at two close but different directions and then finding the corresponding zero position by interpolation.

## 3.4 Hybrid method

The four (or eight) positions method requires to roughly know a priori the True North direction. Moreover, instrument uncertainties (angular sensors and FOG) will cause an error even with an interpolation procedure. Comparatively a hybrid method combining static and dynamic methods ranges the whole circle and performs a measurement at regular interval (e.g. each 10°). At each angular position a 4 positions set of measurement is executed leading to a resulting angular speed given by Eq. (11). A sinus linear least-squares fitting is then applied on the discrete sinus data according to (Rasson, 2009).

There are different ways to implement the hybrid method in the case of Gyrodif. For instance we can choose to perform all measurements with H axis at 90° and then the measurements with H axis at 270°. This would lead to two sine curves. The first one corresponds to sensor UP while the second one is recorded after rotating the H axis by 180 degrees. The resulting phase shift is finally the mean phase of both sinus fitted curves. Another possibility is to take advantage of the static method by performing 4 measurements at each step. Thus only one resulting discrete sinus curve is recorded.



## 4 Results

### 4.1 Interpolated 4-positions method

The interpolated 4 positions method has been tested first. A cost effective FOG has been used for validating the theory. The optimum acquisition time and bias stability have been defined from Allan variance (Fig. 2). They are respectively 500 sec

and 0.05°/h. Two positions around east direction have been arbitrarily defined. The instrument has been installed in the absolute house of Dourbes magnetic observatory. Like conventional DIFlux, GyroDIF has been placed on a geodetic pillar. A "low level" of thermal stability has been established. Room temperature is controlled by means of a standard thermal regulator so that temperature changes are not worse than 2 or 3 degrees peak-peak and an insulated enclosure (10cm thick EPS) has been placed around the device. A series of more than 1800 measurements are presented in Fig. 3.

Standard deviation is about $1\sigma \approx 0.1°$ which is clearly too much compared to international standard. Nevertheless, this dispersion appears to be a white noise so that, when the number of samples is sufficient (here N=1800), the final uncertainty can be reduced to:

$$\sigma_N = \frac{\sigma}{\sqrt{N}} \approx 0.0024° \, , \qquad\qquad (13)$$

Obviously, considering that the pillar is stationary, the mean value is supposed to remain constant during the whole period. This hypothesis may not be encountered in case of in-field deployment. This mean value is also supposed to be unbiased thanks to the measurement protocol. This is the case if and only if we do not take into account the instrument uncertainties and a possible FOG non-linearity, e.g. injection locking or pulling effects (Razavi, 2004). For this last, Eq. 12 suggests that 100 ppm would lead to 20 arcseconds error. Figure 4 presents Dourbes LAMA variometer D0 baseline (Rasson, 2005)

computed from GyroDIF and conventional DIFlux absolute measurements. Both measurements are separated by a small 0.01° offset that would correspond to 3.5nT on the Y component.

The presence of $\theta$ in Eq. (6) and Eq. (11) shows that the North-seeking sensitivity decreases as the latitude increases. Actually, the problem is similar to measuring magnetic declination at high magnetic latitude where the horizontal component is weak. If we consider that automatic observatories are desirable in the Polar regions, testing the sensitivity at high latitude

becomes crucial. This is why a series of measurements has been made in Sodankyla magnetic observatory, latitude 67°22'N. The observed standard deviation in the North-seeking procedure is $1\sigma \approx 0.16°$ which is more than in Dourbes but still manageable. Figure 5 presents the result of interpolated 4-positions measurements in Sodankyla.

### 4.2 Remarks on Absolute Magnetic Declination measurement accuracy

Different sources may contribute to the uncertainty measured in Sect. 4.1. The angular accuracy of AutoDIF and thus GyroDIF is around 6 arcseconds (Poncelet et al., 2017). Both vertical and horizontal angles uncertainties contribute to the



global error. Moreover, this estimated uncertainty is a statistical value computed over a whole turn while the 4 positions method always uses the same positions leading to a systematic error that could be slightly different from the statistical one. In the case of conventional measurements, the observer eyesight and ability to point the target in the same way as a colleague is seldom better than 5 arcecconds and also depends on the telescope optics. Other sources of uncertainty are the pillar

difference; time-synchronization between variometer and absolute instrument, including scalar instrument; and magnetic cleanliness of the absolute room or the observer. It should be noted that intercomparing absolute instruments by performing parallel measurements using a variometer baseline as a yardstick rarely secures accuracies better than +/- 10 arcseconds for Magnetic Declination. The intercomparison session organized during XVII[th] IAGA Workshop on Geomagnetic Observatory instruments, data Acquisition and Processing gives an idea of the usual baseline difference obtained from different couples

instruments/observers. For instance, 25 participants performed a series of absolute measurement on pillar D05 (other participants measured on other pillars). The mean value of each participant series is shown on Fig. 6. Most of the results are within $\pm 2nT$.

## 4.3 Hybrid method

The hybrid method has also been implemented. A 4-positions protocol is executed each 10° starting from 5° to 355° on the

horizontal circle (i.e., around the vertical axis). The whole procedure requires therefore 144 measurements. Figure 7 shows the 36 resulting measurements according to Eq. (11) and the corresponding sinus fitting. In order to keep reasonable measurement duration, FOG signal acquisition time has been reduced to 45 sec per position. Adding the motion time and stabilization time for the bubble level, a whole protocol takes about 2 hours.

The series of measurements presented in Fig. 8 has a standard deviation $1\sigma \approx 0.06°$. Because we may not exclude the possibility that the pillar and the instrument resting on it may change its orientation over the time, we must be able to track this long term angular variation. Therefore a low-pass filter must be implemented. It could be a sliding mean but it is common to use a Kalman filter when working with FOG. In this case, the filtered values have a standard deviation $1\sigma \approx 0.004°$.


The hybrid method has been compared to the conventional measurements (Fig.9). The magnetic (Declination and Inclination) phase has been executed every night between 0h00 and 3h00 UTC while the rest of the time was used for the True North measurement. As for the interpolated 4-position method, comparison is performed on different pillars and the same remarks apply here. Results seem better than in Sect. 4.1 since the difference in Y0 is within $1nT$. However, we should

notice that the number of measurement is limited to 3 weeks. Also only a few comparative conventional measurements have been performed. Nevertheless, as explained in Sect. 4.2, the systematic errors due to angle reading are clearly reduced due to the higher amount of steps.





## 5 Conclusion

In this paper, we presented a new improvement in automation of magnetic observatories. Different methods for automatically finding True North have been established and demonstrated. It appears that the hybrid method is more in accordance with the concept of an automatic setup. Moreover, a series of instrument uncertainties are smoothed during the

sinus fitting step. Results presented here have been obtained with a low cost FOG sensor. A more sensitive device may lead to better and faster result. In particular, high latitude observatories need accurate FOG as $H_e$ then becomes small. Nevertheless, measurements made in Dourbes observatory already meet INTERMAGNET accuracy standards.

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

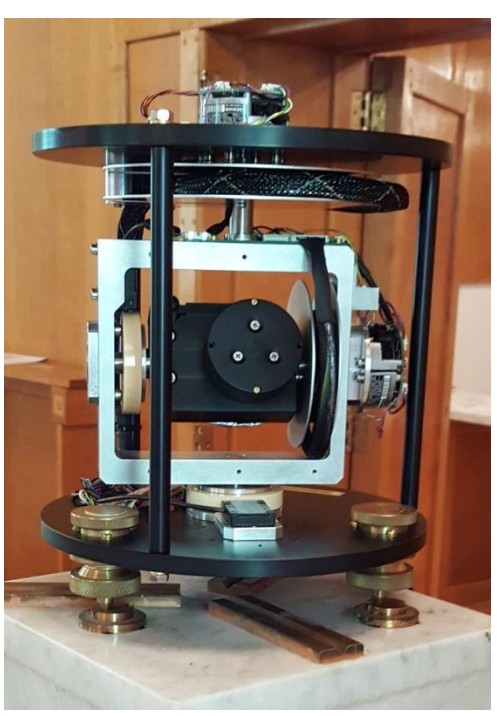

**Figure 1: GyroDIF instrument.**





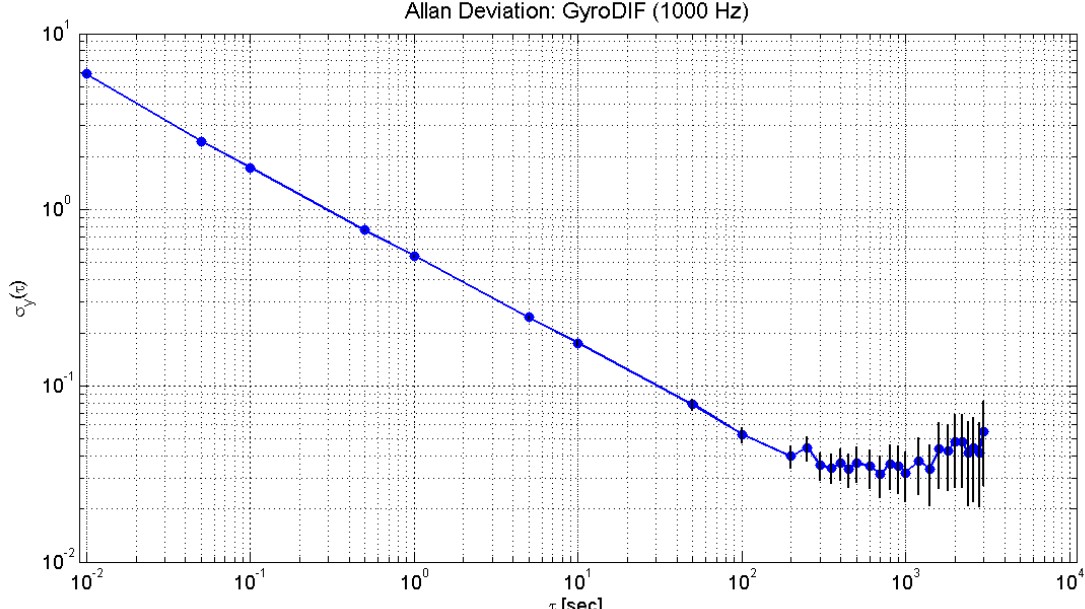

**Figure 2: Allan Variance plot giving the FOG output standard deviation according to the acquisition time. The minimum value gives the bias stability as well as the acquisition optimum time.**

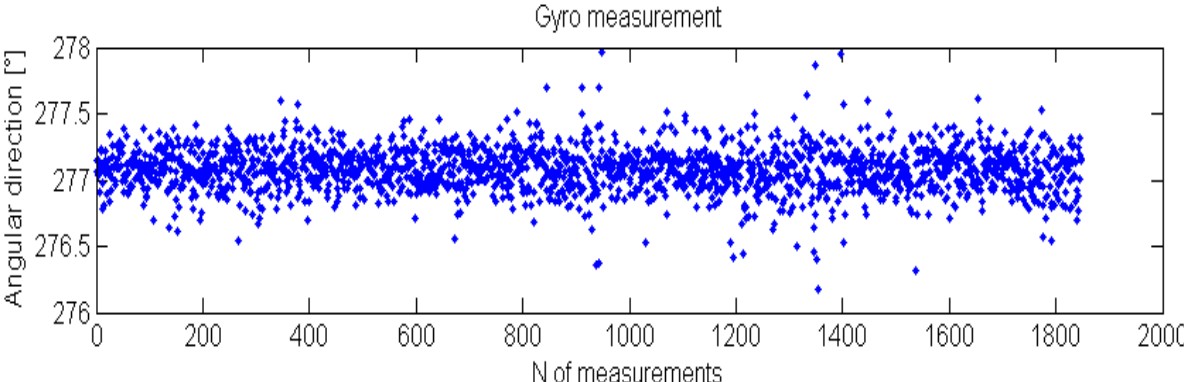

5    **Figure 3: Long terms series of interpolated 4 positions gyro North-seeker measurement (Trace on horizontal circle).**



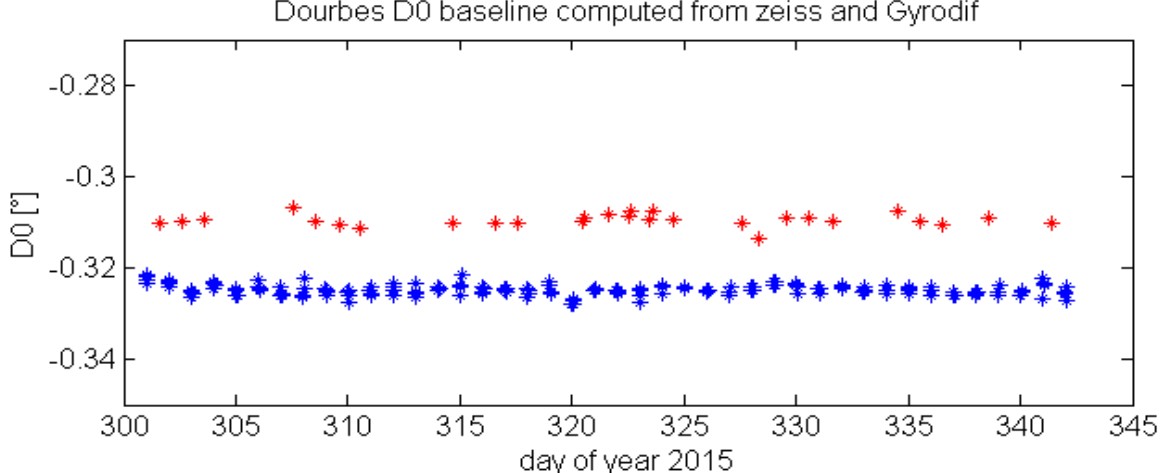

**Figure 4: Baseline D0 comparison. Blue dots are computed from GuroDIF measurements. Red dots are computed from conventional DIFlux (ZEISS 010-B).**

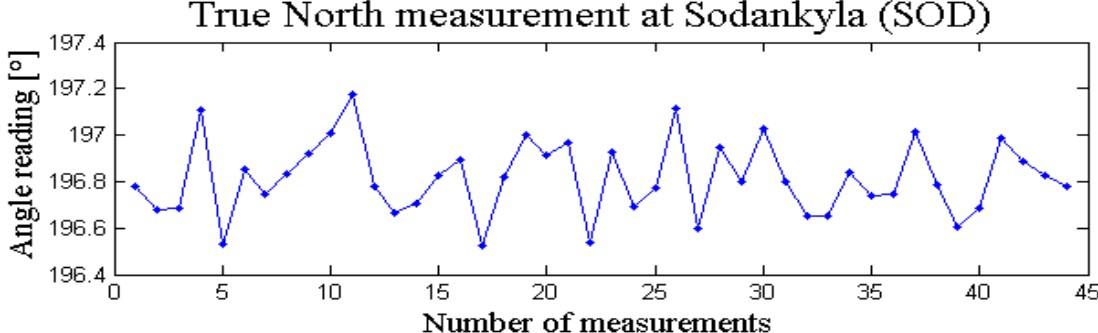

5   **Figure 5: Series of True-North measurements (Trace on horizontal circle) at Sodankyla magnetic observatory. The angle readings correspond to horizontal circle value when instrument is pointing True-North.**

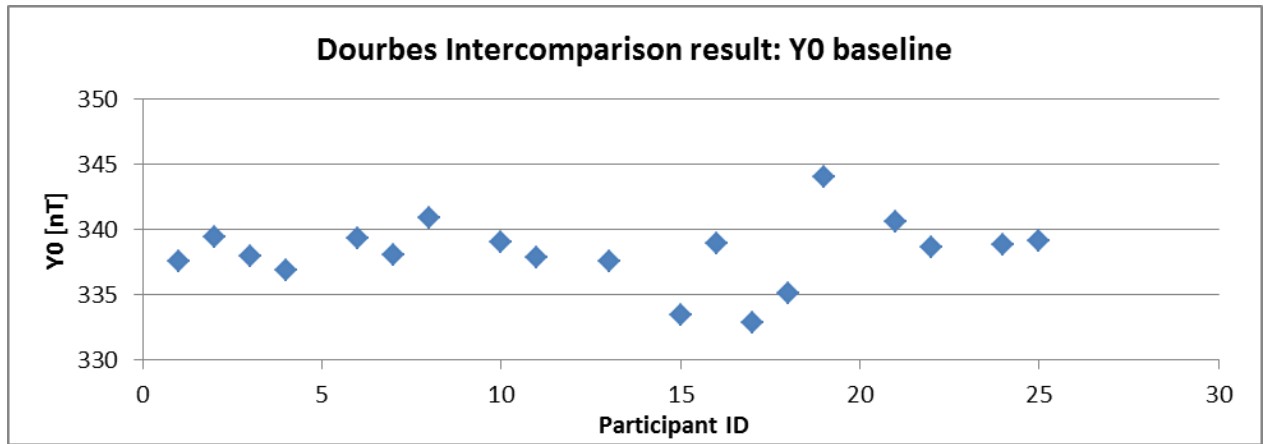

**Figure 6: Result of the intercomparison session organized during the XVII<sup>th</sup> IAGA Workshop on Geomagnetic Observatory instruments, data Acquisition and Processing. Each value corresponds to the mean result of an observer/instrument series performed on pillar D05. East component Y0 is shown.**

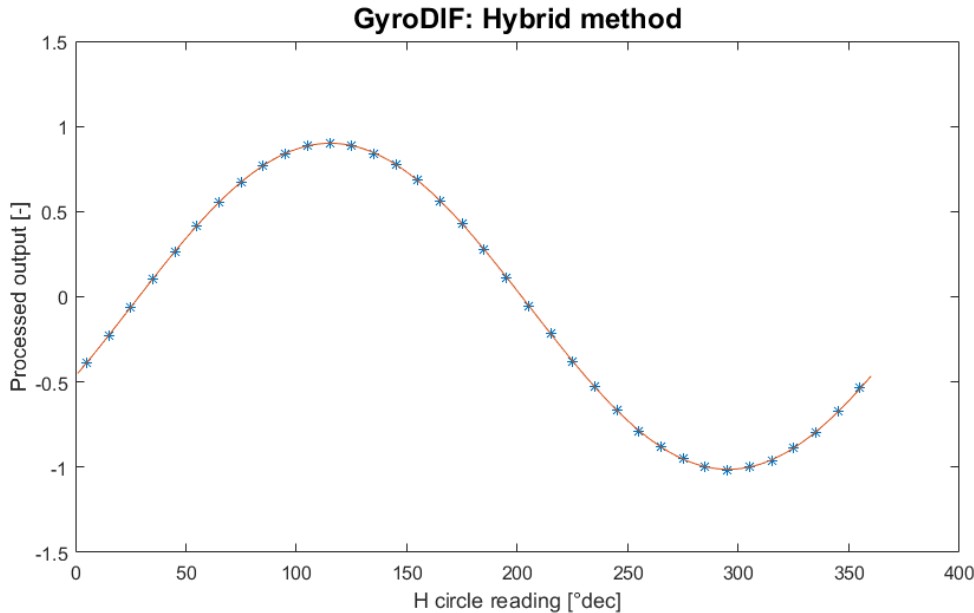

**Figure 7: Fibre Optic Gyro output signal due to Earth rotation when its sensitive axis scans the horizontal plane in Dourbes. The maximum of the sine function corresponds to True North. Blue: Hybrid method $\omega_r$ according Eq. (11). Red: sinus fitting.**





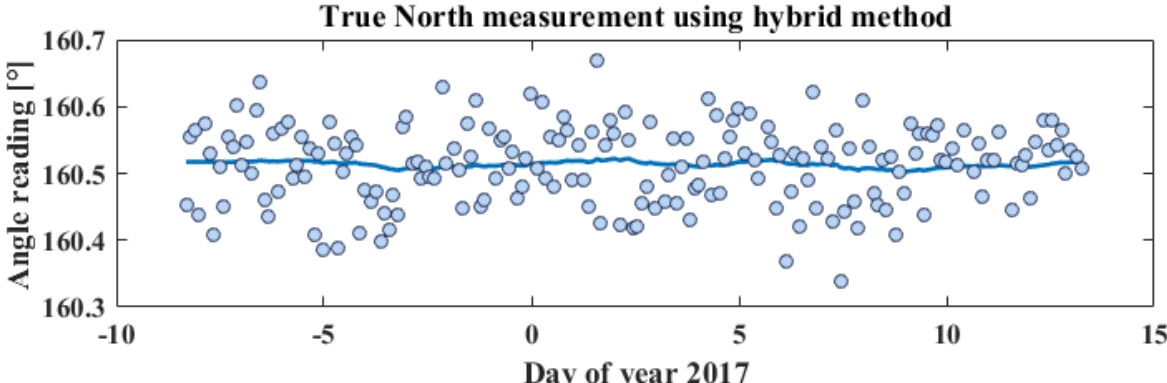

**Figure 8: Series of True North measurements (trace on horizontal circle) obtained by means of hybrid method (Dots). The solid line corresponds to the true North determination after passing through a Kalman filter.**

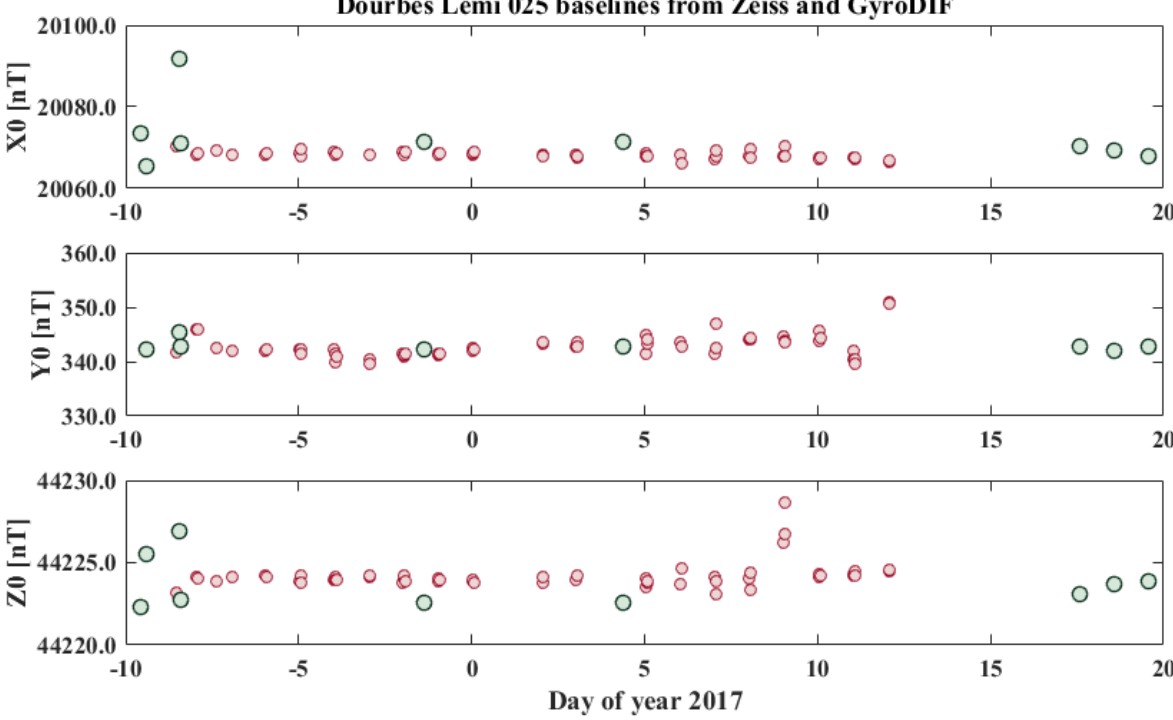

5  **Figure 9: Blue: Dourbes LEMI 025 Baselines computed from GyroDIF measurements (Red). The True North direction used in the Y0 baseline is determined by means of hybrid method. The green dots are computed from conventional DIFlux measurements.**