# Peer review of "Automatic True North detection during absolute magnetic declination measurement"

_Geoscientific Instrumentation, Methods and Data Systems, 2017_

## Referee Comment (RC1) · H.-P. Brunke (Referee) · 10 Jul 2017

Review of "**Automatic True North detection during absolute magnetic declination measurement**" by Gonsette et.al.

In this manuscript a new system using an optical gyroscope for north detection is presented. North detection is an essential part of called "Absolute Measurements" at geomagnetic observatories. Previous work to do these "Absolutes" in an automated way are considered and the relevance of the presented new system in this framework is pointed out. The new system is of use, if a stable azimuth marks as reference to the geographic coordinate system is not available.

As I already stated in my short review with respect to the initial submission **I think the manuscript is clearly suitable for publication.**

**But nevertheless I have two minor remarks.**

1.) **Title:** Would it not be good to mention already in the title, that a fiber optic gyroscope (FOG) is used? The current title could suggest at a first glance, that the true north is to be determined **by** absolute magnet measurement. Mentioning the method FOG could perhaps attract more interested persons.

2.)**Page 3, line 5.** Do you really need to bother the central limit theorem? The central limit theorem (CLT) establishes that, when independent random variables are added, their sum tends toward a normal distribution even if the original variables themselves are not normally distributed. Perhaps: "The previous equation suggests increasing the sampling time, in order to increase the realizations of $\phi$ measurements for a statistically firmer result."

---

## Referee Comment (RC2) · C. Turbitt (Referee) · 17 Jul 2017

One minor correction: Capitalise the document title on Page 10, line 8. Otherwise accept as is. Thanks to the authors for taking on board previous comments.
* * *

---

## Author Comment (AC1) · 8 Aug 2017

1.) Title: Would it not be good to mention already in the title, that a fiber optic gyroscope (FOG) is used? The current title could suggest at a first glance, that the true north is to be determined by absolute magnet measurement. Mentioning the method FOG could perhaps attract more interested persons.

The method is not limited to the FOG technology. RLG, HRG and Mems could also be used (if they are accurate enough and available for non military users). However, it is true that "FOG based" in the title could be attractive. The title could be adpted as follow: "FOG-based automatic True North detection for absolute magnetic declination measurment".

[Figure]

2.)Page 3, line 5. Do you really need to bother the central limit theorem? The central limit theorem (CLT) establishes that, when independent random variables are added, their sum tends toward a normal distribution even if the original variables themselves are not normally distributed. Perhaps: "The previous equation suggests increasing the sampling time, in order to increase the realizations of phi measurements for a statistically firmer result."

Maybe just: "The previous equation suggests to increase the sampling time in order to reduce the white noise. However, the bias is subject to ..."

---

## Author Comment (AC2) · 8 Aug 2017

One minor correction: Capitalise the document title on Page 10, line 8. Otherwise accept as is. Thanks to the authors for taking on board previous comments.

I will change as follow:

St-Louis, B.J.: INTERMAGNET Technical Reference Manual, V4.6, INTERMAGNET, http://www.intermagnet.org/publication-software/technicalsoft-eng.php, 2012

However the guidelines for authors suggests:

Titles and headings follow sentence-style capitalization (i.e. first word and proper nouns only).

So I suppose that it will be corrected during the proofreading step.